# AQ-Bench: A Benchmark Dataset for Machine Learning on Global Air Quality Metrics

Clara Betancourt[1], Timo Stomberg[1,2], Ribana Roscher[2], Martin G. Schultz[1], and Scarlet Stadtler[1]

[1]Jülich Supercomputing Centre, Jülich Research Centre, Wilhelm-Johnen-Straße, 52425 Jülich, Germany
[2]Institute of Geodesy and Geoinformation, University of Bonn, Nußallee 17, 53115 Bonn, Germany

**Correspondence:** Martin G. Schultz (m.schultz@fz-juelich.de)

**Abstract.** With the AQ-Bench dataset, we contribute to the recent developments towards shared data usage and machine learning methods in the field of environmental science. The dataset presented here enables researchers to relate global air quality metrics to easy-access metadata and to explore different machine learning methods for obtaining estimates of air quality based on this metadata. AQ-Bench contains a unique collection of aggregated air quality data from the years 2010-2014 and metadata at more than 5500 air quality monitoring stations all over the world, provided by the first Tropospheric Ozone Assessment Report (TOAR). It focuses in particular on metrics of tropospheric ozone, which has a detrimental effect on climate, human morbidity and mortality, as well as crop yields. The purpose of this dataset is to produce estimates of various long-term ozone metrics based on time-independent local site conditions. We combine this task with a suitable evaluation metric. Baseline scores obtained from a linear regression method, a fully connected neural network and random forest are provided for reference and validation. AQ-Bench offers a low-threshold entrance for all machine learners with an interest in environmental science and for atmospheric scientists who are interested in applying machine learning techniques. It enables them to start with a real-world problem relevant to humans and nature. The dataset and introductory machine learning code are available at https://doi.org/10.23728/b2share.30d42b5a87344e82855a486bf2123e9f (Betancourt et al., 2020) and https://gitlab.version.fz-juelich.de/esde/machine-learning/aq-bench. AQ-Bench thus provides a blueprint for environmental benchmark datasets as well as an example for data re-use according to the FAIR principles.

**Graphical abstract**

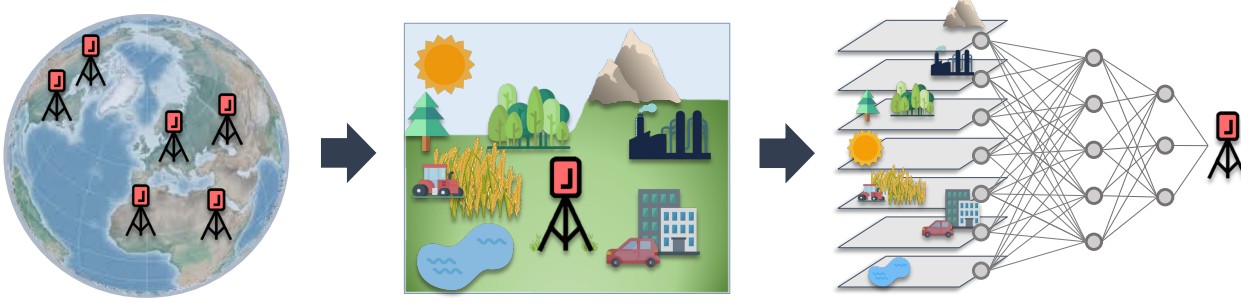

The AQ-Bench dataset contains long-term air quality metrics and metadata at sites around the globe. Map by GMT (Wessel et al., 2016)

The air quality at a site is influenced by its surroundings.

The proposed machine learning task is to train a machine learning algorithm which maps from metadata to long-term air quality metrics at measurement sites.

## 1 Introduction

In recent years, machine learning has achieved remarkable success in areas such as pattern-, image- and speech- recognition by usage of increasing computing power, innovative algorithms and high data availability (Krizhevsky et al., 2012; Amodei et al., 2016; Silver et al., 2016). This aroused the interest of environmental scientists to explore the application of machine learning and data-driven methods in their fields. The strength to be exploited is the ability of machine learning algorithms to find complex relationships in large multivariate, inhomogeneous datasets (as described in Wise and Comrie, 2005; Porter et al., 2015, e.g.).

In air quality research, there is one pollutant which is especially challenging to track: tropospheric ozone, a toxic trace gas which harms human health, vegetation and also impacts the climate (Cooper et al., 2014; Monks et al., 2015). Tropospheric ozone is difficult to track because it has no direct emission sources, but is produced as a secondary air-borne pollutant by several chemical reaction chains involving a large variety of precursors and photochemistry. With a lifetime of days to weeks (Wallace and Hobbs, 2006), the ozone concentration is affected by various physical and chemical processes which produce and destroy ozone. Therefore, ozone is a scientifically interesting candidate for machine learning applications: It is influenced by many interconnected environmental factors - and it is interesting to see if machine learning algorithms can learn these.

Data-driven atmospheric chemistry research was combined with machine learning from the late 1990s, to model and predict surface ozone concentrations in an alternative way to multivariate regression from (Yi and Prybutok, 1996; Comrie, 1997; Elkamel et al., 2001; Caselli et al., 2009). These data driven approaches take ground based measurements as input and predict the pollutant concentrations for the next days at individual locations. The principle behind recent machine learning applications in ozone research is often a similar principle as Schultz et al. (2020) described for weather data: The input data are directly mapped to a specific data product, e.g. from meteorological and past ozone measurements to the next day's maximum ozone

value. In recent studies, Sayeed et al. (2020) and Kleinert et al. (2020) predicted regional ozone time series with convolutional neural networks and meteorological input data. Furthermore, Silva et al. (2019) trained a feed forward neural network to output ozone dry deposition at two forest measurement sites. Moreover, within computationally complex components of atmospheric chemistry models, machine learning techniques are used as emulators or surrogate models. They replace for example costly atmospheric chemistry and micro-physical calculations to improve computational performance of the models (Kelp et al., 2020). In addition, machine learning is applied in the calibration of low-cost sensors for air quality measurements in order to account for the diverse sources of interference with these measurements (Schmitz et al., 2021; Wang et al., 2020). Nevertheless, to our knowledge there are currently no machine learning projects that attempt analyzing and predicting ozone on the global scale, for longer time periods and with many kinds of metadata.

Developments in machine learning are accelerated by the existence of precompiled benchmark datasets, that allow machine learners to try out specific tasks, exchange solutions and compete with each other (LeCun et al., 2010; Deng et al., 2009; Rasp et al., 2020). Benchmarks can also be used for the development of explainable artificial intelligence approaches (Kierdorf et al., 2020; Roscher et al., 2020). So far, few of such benchmark datasets exist in the field of environmental science, especially related to air quality. While air quality data are in principle easily accessible from a variety of archives, there is often incomplete information and insufficient metadata to develop useful machine learning applications from this data. Furthermore, harmonization of such data from different sources, which is needed to achieve a global picture of ozone air pollution, is a difficult and time-consuming task.

With the AQ-Bench dataset, we aim to fill this gap and provide a dataset of global long-term air quality metrics and metadata compiled from the TOAR database (Tropospheric Ozone Assessment Report, Schultz et al. (2017)). To make these data usable for machine learning developments, this paper also describes the specific task of mapping between the metadata and the air quality metrics. Our ready-to-use, fully documented dataset is freely available under the DOI  https://doi.org/10.23728/b2share.30d42b5a87344e82855a486bf2123e9f (Betancourt et al., 2020). We also provide our baseline machine learning code at https://gitlab.version.fz-juelich.de/esde/machine-learning/aq-bench, offering a low-threshold entrance to machine learning in environmental science within a relevant research topic. In Sect. 2 of this paper we present the main factors affecting tropospheric ozone as the scientific background for the design of the AQ-Bench data set. Section 3 introduces the TOAR data products from which AQ-Bench was constructed. In Sect. 4, we describe the dataset itself. Section 5 contains the machine learning task for AQ-Bench and three baseline experiments to evaluate the applicability of these data in the machine learning context. We discuss opportunities and challenges of AQ-Bench, and give problem-related expected difficulties in Sect. 6. Information on data and code availability is given in Sect. 7, followed by a conclusion in Sect. 8.

## 2    What factors influence ozone?

Ozone ($O_3$) is a toxic greenhouse gas. While stratospheric ozone protects life on the planet's surface from ultraviolet radiation, tropospheric ozone is detrimental to human health, vegetation and climate. The AQ-Bench dataset and this paper focus exclusively on tropospheric ozone, more precisely the near-surface ozone to which humans, animals and plants are exposed. Ozone

is a secondary pollutant that is formed from emissions of precursor substances and undergoes a variety of physical and chemical processes during its atmospheric lifetime. Figure 1 summarizes these processes, and is further elaborated in the following Subsections. How the described processes translate into the data in AQ-Bench, is described in the dataset description (Sect. 4).

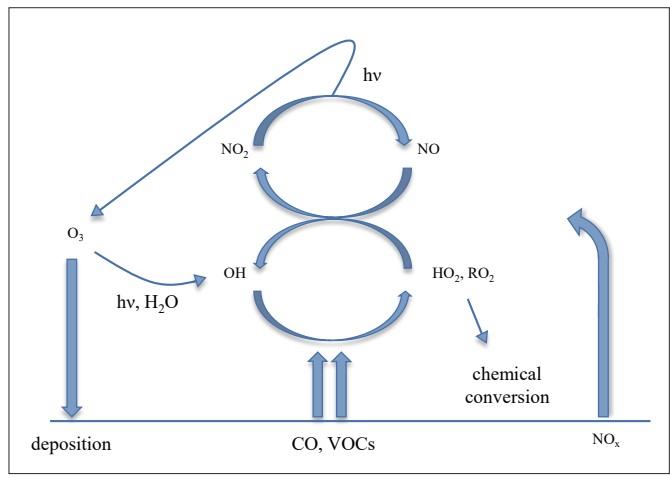

**Figure 1.** Simplified scheme describing the ozone chemical cycle. Figure adapted and modified from Jacob (2000). See text for elaboration.

## 2.1 Precursor emissions

The most important ozone precursors are nitrogen oxides, carbon monoxide and volatile organic compounds (denoted as $NO_x$, CO and VOCs in Figure 1; note that $NO_x = NO_2 + NO$). Many of these precursors are emitted by human activities, e.g. from traffic, industry and agriculture (Benkovitz et al., 1996; Field et al., 1992). $NO_x$ concentrations resulting primarily from combustion processes are especially high at very heavily polluted sites such as in city centers or near power plants. Industrial and traffic pollution are closely related to energy consumption depending on population density and economical activities. Agriculture machinery emits similar trace gases as traffic or industry. Moreover, agricultural plants are often fertilized, which adds more trace gas emissions (Veldkamp and Keller, 1997). In addition to emissions from human activities, several processes in nature also lead to emissions, especially of VOC compounds. For example, plants emit VOCs which are often more reactive (and could therefore produce more ozone) than VOCs emitted from human activities. The exact emission patterns vary among the types of plants and are thus related to land cover. Agricultural fields, forests and grasslands therefore yield different magnitudes and seasonal cycles of VOC emissions (Simpson et al., 1999). Emissions can also occur from oceans, barren land and snow or ice covered surfaces. For example, the latter emit substantial quantities of $NO_x$ in Arctic regions (Wang et al., 2007).

## 2.2 Ozone chemistry

The daily average ozone volume mixing ratios vary in the orders of magnitude from 10 to 100 ppbv (parts per billion by volume), with a lifetime of days to weeks (Wallace and Hobbs, 2006). Ozone has practically no direct emissions but is exclusively formed through atmospheric chemical reactions. The chemical processes leading to ozone formation are driven by ultraviolet radiation (denoted with $h\nu$ in Fig. 1). At wavelengths $< 0.43$ nm, photons convey enough energy to release chemical bonds in nitrogen dioxide ($NO_2$) molecules. This process (photo dissociation) leads to the formation of nitrogen oxide (NO) and a free oxygen radical (O). NO is also a radical and thus recombines quickly, while O collides with a high probability with $O_2$ and forms $O_3$. The produced $O_3$ is removed rapidly when it reacts with NO to $NO_2 + O_2$. The reactions form a null cycle, because $O_3$ is both created and destroyed. The cycle stabilizes at a certain $O_3$ concentration, depending on the available $NO_2$, ultraviolet light intensity and temperature. Up to a certain point, the ozone concentration rises with increasing $NO_2$ concentrations.

The dynamic equilibrium of this cycle can be altered by the presence of VOCs and CO (denoted as primary emissions in Fig. 1), which provide chemical pathways to convert NO to $NO_2$ without the destruction of $O_3$ by oxidation (oxidized pollutants denoted as $HO_2$ and $RO_2$ in Fig. 1). This leads to a non-linear system, where $O_3$ concentrations depend on the ratio of VOCs + CO and $NO_x$ ($= NO + NO_2$) concentrations. During daytime, $O_3$ can photo dissociate and recombine with water vapor ($H_2O$ in Fig. 1), thereby forming hydroxy radicals (OH in Fig. 2) which fuel a large share of atmospheric oxidation. There are several thousand chemical reactions occurring in the atmosphere, which need to be considered for an adequate description of ozone formation and loss processes and Fig. 1 only provides a very small glimpse on this rather complex system. For more details on ozone chemistry we refer to Brasseur et al. (1999).

## 2.3 Transport and loss processes

During its atmospheric lifetime, $O_3$ can be transported on spatial scales of hundreds or even thousands of kilometers (Schultz et al., 1999), until it is removed via atmospheric chemical reactions and deposition (indicated with downwards pointing arrows in Fig. 1). Primary chemical loss of $O_3$ is rather indirect via removal of $NO_2$ in polluted regimes and radical-radical reactions in clean environments with low $NO_2$ concentrations. Besides the chemical loss, $O_3$ can be removed by deposition on surfaces, especially on the leaves of natural or agricultural plants (Emberson et al., 2000). Ozone irreversibly damages plant tissue when the plant leaves take it up (Schraudner et al., 1997), leading to reduced crop yields (Mills et al., 2011). Ozone deposition on water surfaces is relatively slow, but due to the large extent of them, this process also matters in the context of the global ozone budget (Luhar et al., 2018).

## 2.4 Interconnected factors

In the following, we describe how the influences of ozone precursor emission, chemistry, transport and loss (described in Sects. 2.1 - 2.3) can come together. The combination of chemistry and transport of air pollutants favors ozone formation downwind of sites with high precursor exhaust. A typical example are summertime rural areas downwind of larger city centers,

where peak ozone values can often be observed (Xu et al., 2011). In the close vicinity of power plants or in city centers, $NO_x$ is often very high and low ozone levels are observed (Sillman, 1999).

There are several geographical factors which determine the rates of chemical formation and loss of ozone. These factors can result in different mixes of ozone precursor emissions, varying reaction rates and varying rates of deposition. For example, the climate in a certain location determines the vegetation cover and the local weather. Since temperatures near the equator are high and more intense sunlight is available, ozone levels are generally higher there than near the poles. Moreover, at higher altitudes the air is generally cooler and drier, which leads to changes in reaction rates. Local flow patterns can also influence

the ozone concentration, for example through the transport of air masses from valley to mountain tops (Kaiser et al., 2007).

        Besides natural geographic factors, political decisions can also influence ozone formation. Many governments and decision makers worldwide strive to reduce air pollution by emission regulation, but these regulations differ between countries and may be implemented with more or less rigor. Ozone regulation is more difficult than that of primary air pollutants as one has to limit both VOC and $NO_x$ emissions in order to control ozone, because of the chemical cycles described in Sect. 2.2.

Although ozone has a rather long lifetime, the local ozone concentration can change substantially in a matter of minutes and on scales of meters (e.g. in a street canyon), but it can also remain stable across hundreds of kilometers and for several weeks (e.g. at higher altitudes over the oceans).  The "radius of influence" within which ozone is determined by nearby precursor emissions and deposition surfaces is typically about 25 km in mid-latitude areas (European Union, 2008). All in all, ozone concentrations measured at a station are determined by many interconnected influences from precursor emissions, land

use / land cover and the local weather conditions. Many of these factors are poorly quantified and often the interconnections are not understood well yet (Schultz et al., 2017). With AQ-Bench and the machine learning task described below we want to explore a novel way of using a multitude of geographical features to predict ground-level ozone around the world. The details of data selection are described in Sect. 4, while the machine learning task is provided in Sect. 5.1.

## 3   TOAR data products

The TOAR database (Schultz et al., 2017) was created in context of the Tropospheric Ozone Assessment Report (TOAR). It contains one of the world's largest collections of near-surface ozone measurements, gathered from public bodies, research institutions and air quality networks all over the world. TOAR data products enabled the first comprehensive global assessment of the tropospheric ozone distribution and trends  (Schultz et al., 2017; Fleming et al., 2018; Gaudel et al., 2018; Lefohn et al., 2018; Chang et al., 2017; Young et al., 2018; Mills et al., 2018; Tarasick et al., 2019; Xu et al., 2020). In the spirit of FAIR data

usage (Wilkinson et al., 2016), these data products are openly available via the JOIN graphical interface[1], a REST interface[2], and through the PANGAEA repository[3].

---

[1]https://join.fz-juelich.de/
[2]https://join.fz-juelich.de/services/rest/surfacedata/
[3]https://doi.org/10.1594/PANGAEA.876108

For the AQ-Bench dataset we selected and harmonized air quality metrics and metadata from TOAR (see Sect. 4 and Appendix C). This section therefore contains a description of these selected data products, introducing the concepts of metrics and metadata.

## 3.1 Air quality metrics

The TOAR database contains hourly ozone measurements, transmitted from air quality observation sites. The data providers conduct quality control on these data by calibrating the measurement devices and setting suitable instrument parameters. In a second step of data curation, the TOAR database administrators conduct a statistical analysis of the data to identify and remove low-quality data (Schultz et al., 2017). Hourly data are usually aggregated into statistics or "metrics" for further analysis. Ozone metrics consolidate air quality properties of longer time series (e.g. a season or a year) in a single figure, which can then be directly used for a scientific assessment and in decision making. Longer aggregation periods also average out short term weather fluctuations. There are specific metrics for different areas of ozone impact assessments (respiratory and cardiovascular disease, vegetation damage, climate impacts) and control.

The JOIN web-service is connected to the TOAR database and provides more than 30 of the most frequently used metrics as data products, calculated on-demand from hourly data. Besides these specialized metrics, also basic statistics such as average, median and percentiles are available in JOIN. In the context of evaluating air quality, the validity of reported ozone metrics hinges on the data capture. Typically, statistical aggregations (i.e. metrics) of air quality data can only be used for decisions on attainment or non-attainment of air quality standards, if at least 75 % of the (hourly) samples in a dataset were reported. In this sense, the validity of ozone metrics is tied to the data completeness and we will use the term "valid data" to indicate samples with sufficient coverage of accurate data. All metrics which are part of AQ-Bench, are listed in Table 2 of the next section. Documentation and further information on all available metrics including data capture criteria are available in Schultz et al. (2017) and Lefohn et al. (2018).

## 3.2 Station metadata

The TOAR database also contains geographical information on air quality measurement station locations, i.e. station metadata. Metadata gives background information on the measurement site, where the data was retrieved from, and thus enables to characterize the location. These metadata are collected from different sources. Some data, for instance station coordinates and altitude are given by the data providers and quality controlled by TOAR. Others were derived from data sources with individual quality control, such as satellite earth observations. For a complete list of the available metadata attributes see Schultz et al. (2017) and the REST interface[2].

For the AQ-Bench dataset described in this paper, we selected metadata from the TOAR database which characterize measurement locations and their surroundings with respect to pollution-relevant properties as introduced in Sect. 2. They are listed in Table 1 of the next Section.

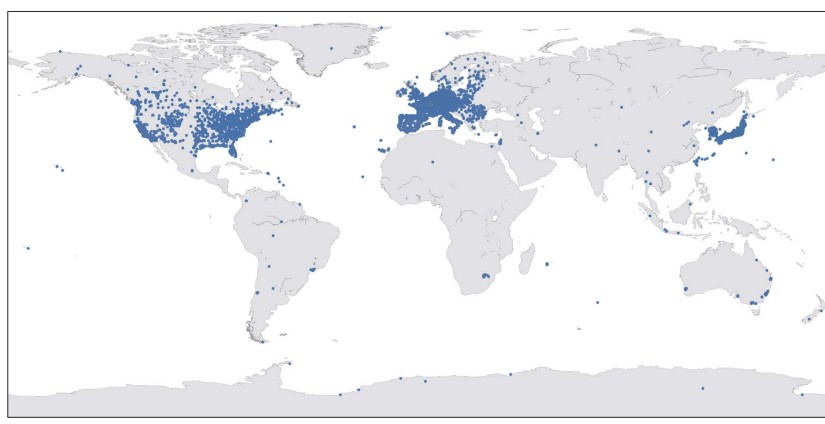

**Figure 2.** Worldwide measurement stations which are part of AQ-Bench, selected from the TOAR database. Map by GMT (Wessel et al., 2019).

## 4  AQ-Bench dataset description

The AQ-Bench dataset consists of metadata and aggregated ozone metrics from the years 2010-2014 at 5577 measurement
stations all over the world, compiled from the TOAR database. The point of interest is to determine the resulting ozone metrics (see Sect. 3.1) given all environmental influences (Sect. 2) represented by metadata (Sect. 3.2). Our contribution in data preparation is to pick metadata with expert knowledge, relate it to processes, and aggregate air quality data to metrics in a way that it is representative for long time periods and meaningful in a machine learning context.

Three key points in the conception of this benchmark dataset are: 1) As targets, we use aggregated air quality metrics over
five years. These are not influenced by short-term weather and emission forcings, but by site conditions on the climatological time scale. 2) Many known environmental influences on ozone are on short time scales (see Sect. 2), but we aim to predict long-term air quality conditions at the sites. Thus, we have identified which station metadata are the climatological representations of these short forcings. 3) We use a - to our knowledge unprecedented - variety of metadata that contains diverse information about environmental influences on the climatological scale. These metadata are sometimes not directly descriptive of the influences,
but rather proxies for them. The benefits of machine learning must be taken to relate these proxies to air quality metrics.

This aggregated, climatological approach makes it possible to cover air quality data over a long period of time on the global scale with a relatively small and compact dataset. Yet, aggregated data accounts for long-term air quality conditions at a site, and daily or hourly influence on ozone variations not considered. Figure 2 gives an overview of all TOAR air quality monitoring stations included in AQ-Bench.

## 4.1 Station metadata

A summary of metadata in AQ-Bench is given in Table 1. The data originates from the TOAR database (Sect. 3) See Appendices A and C for details on the data sources and harmonization for machine learning purposes. The metadata contains proxies for environmental influences on ozone on the climatological scale. In the following, we give two examples.

As mentioned in Sect. 2, ozone is influenced by weather. Likewise, ozone on longer time scales is influenced by climate. One variable in the AQ-Bench dataset is the *climatic zone* in which the site is located. The climate zone provides simplified information about climatic conditions at a location, for example, whether it is hot or cold, humid or dry, or of tropical climate.

A second example are ozone precursor emissions. In Sect. 2.1 we outlined that they are emitted by, for example, traffic and human activities. This means that the *population density* at a site is a good proxy for these activities. A second - more subtle - proxy is the *stable nightlight* at a location. It is the average intensity of light during night as seen from space, an indicator for industrial activity. In Sect. 2.2, we pointed out that ozone is often formed downwind of sites with high human and industrial activity. Therefore, in the AQ-Bench dataset, we do not only give *population density* and *stable night lights* at a site, but also related statistics of the closer surroundings. One example is the maximum *population density* in a radius of 5 km around the station.

All variables of the AQ-Bench dataset can be related to environmental impacts on the climatological time scale. We indicate the proxies in the right column of Table 1. Machine learning can make use of these proxies, even if they are not directly related to ozone concentrations.

Table 1: The station metadata of AQ-Bench.

| Variable | Unit | Type | Proxy for... |
| --- | --- | --- | --- |
| Country | - | categorical | Emission regulation |
| HTAP region | - | categorical | World region set by the Task Force on Hemispheric Transport of Air Pollution http://htap.org |
| Climatic zone | - | categorical | Temperature, humidity, radiation |
| Longitude | deg | circular | - |
| Latitude | deg | continuous | Radiation, temperature |
| Altitude | m | continuous | Sinks, temperature |
| Relative altitude | m | continuous | Local flow patterns |
| Type | - | categorical | Industry / traffic emissions |
| Type of area | - | categorical | Proximity to human settlement |
| Water in 25km area | % | continuous | Deposition |
| Evergreen needle leaf forest in 25 km area | % | continuous | VOC Emissions, deposition |

*(continued on next page)*

| Variable | Unit | Type | Proxy for... |
|---|---|---|---|
| Evergreen broadleaf forest in 25 km area | % | continuous | VOC Emissions, deposition |
| Deciduous needle leaf forest in 25 km area | % | continuous | VOC Emissions, deposition |
| Deciduous broadleaf forest in 25 km area | % | continuous | VOC Emissions, deposition |
| Mixed forest in 25 km area | % | continuous | VOC Emissions, deposition |
| Closed shrub lands in 25 km area | % | continuous | VOC Emissions, deposition |
| Open shrub lands in 25 km area | % | continuous | VOC Emissions, deposition |
| Woody savannas in 25 km area | % | continuous | VOC Emissions, deposition |
| Savannas in 25 km area | % | continuous | VOC Emissions, deposition |
| Grasslands in 25 km area | % | continuous | VOC Emissions, deposition |
| Permanent wetlands in 25 km area | % | continuous | VOC Emissions, deposition |
| Croplands in 25 km area | % | continuous | Agricultural emissions |
| Urban And Built-Up in 25 km area | % | continuous | Human settlement |
| Cropland / Natural vegetation mosaic in 25 km area | % | continuous | Emissions, agriculture, deposition |
| Snow and ice in 25 km area | % | continuous | Factor on ozone formation |
| Barren or sparsely vegetated in 25 km area | % | continuous | Emissions, deposition |
| Wheat production | 1000 Tons | continuous | Agricultural emissions |
| Rice production | 1000 Tons | continuous | Agricultural emissions |
| $NO_x$ emissions | $g\,m^{-2}\,y^{-1}$ | continuous | $NO_x$ emissions |
| $NO_2$ full column | $10^5\ molec\,cm^{-2}$ | continuous | $NO_2$ |
| Population density | $person\,km^{-2}$ | continuous | Human emissions |
| Max population density 5km | $person\,km^{-2}$ | continuous | Human emissions near-by |
| Max population density 25km | $person\,km^{-2}$ | continuous | Human emissions in area of influence |
| Nightlight 1 km | brightness index | continuous | Industrial activity |
| Nightlight 5 km | brightness index | continuous | Industrial activity near-by |
| Max nightlight 25 km | brightness index | continuous | Industrial activity in area of influence |

## 4.2 Ozone metrics

The AQ-Bench dataset contains annually aggregated, averaged (years 2010-2014) ozone metrics as introduced in Sect. 3.1.
There are therefore two steps involved in obtaining the metrics: 1) Getting up to five yearly metrics between 2010-2014 from
hourly measurements, including data cover criteria to validate the metrics 2) averaging over these five years. If less than
two yearly values are available, the value is considered missing. Missing values are denoted with -999 in the dataset. Some

suspiciously high values were sorted out, as documented in Appendix C. A summary of all metrics and their data capture criteria is given in Table 2. More details on the process of ensuring robustness through data capture are given in Appendix B.

Table 2: The ozone metrics of AQ-Bench. The unit is ppb (= parts per billion) for all metrics except the nvgt metrics, where it is the number of days.

| Metric | Description | Relevant field |
|---|---|---|
| Average values | Annual average value. No data capture criterion is applied, i.e. an average is valid if at least one hourly value is present. | Basic statistics |
| Daytime average | "Daytime average" is defined as average of hourly values for the 12-h period from 08:00 to 19:59 solar time. All hourly values in the aggregation period are averaged, and the resulting value is valid if at least 75 % of hourly values are present. | Basic statistics |
| Nighttime average | Same as "Daytime average" but accumulated over the daily interval from 20:00 to 07:59 solar time. | Basic statistics |
| Median | Median daily mixing ratio over one year. At least 10 hourly values must be present to accept a daily median value as valid. | Basic statistics |
| 25 % percentile | 25th-percentile of daily values in one year. At least 10 hourly values must be present to accept a daily percentile value as valid. | Basic statistics |
| 75 % percentile | As "25 % percentile", but for the 75th-percentile. | Basic statistics |
| 90 % percentile | As "25 % percentile", but for the 90th-percentile. | Basic statistics |
| 98 % percentile | As "25 % percentile", but for the 98th-percentile. | Basic statistics |
| dma8eu | Daily maximum 8-hour average statistics according to the EU definition. 8-hour averages are calculated for 24 bins starting at 17:00 local time of the previous day. The 8-h running mean for a particular hour is calculated on the concentration for that hour plus the following 7 hours. If less than 75% of the data are present (i.e. less than 6 hours), the average is considered missing. For annual aggregation, the 26th highest daily 8-hour maximum of the aggregation period will be computed. Note that in contrast to the official EU definition, a daily value is considered valid if at least one 8-hour average is present. | Human health |
| avgdma8epax | Average value of the daily "dma8epax" statistics during the aggregation period. "dma8epax" is the same as "dma8eu", but hourly bins start at 00:00 instead of 17:00. | Human health |

*(continued on next page)*

| Metric | Description | Relevant field |
|---|---|---|
| drmdmax1h | Maximum of the 3-months running mean of daily maximum 1-hour mixing ratios during the aggregation period of one year. | Human health |
| W90 | Daily maximum W90 5-h Experimental Exposure Index: EI = SUM($w_i C_i$) with weight $w_i = 1 / [1 + M \exp(\text{-A } C_i/1000)]$, where M = 1400, A = 90, and where $C_i$ is the hourly average $O_3$ mixing ratio in units of ppb. For each day, 24 W90 indices are computed as 5-hour sums, requiring that at least 4 of the 5 hours are present (75%). If a sample consists of only 4 data points, a fifth value shall be constructed from averaging the 4 present mixing ratios. For annual aggregation, the 4th highest W90 value is computed, but only if at least 75% of days in this period have valid W90 values. | Vegetation |
| AOT40 | Daily 12-h AOT40 values are accumulated using hourly values for the 12-h period from 08:00 until 19:59 solar time interval. AOT40 is defined as cumulative ozone above 40 ppb. If less than 75% of hourly values (i.e. less than 9 out of 12 hours) are present, the cumulative AOT40 is considered missing. When there exist 75% or greater data capture in the daily 12-h window, the scaling by fractional data capture ($n_{total}/n_{present}$) is utilized. For annual statistics, the daily AOT40 values are accumulated over the aggregation period and scaled by ($n_{total}/n_{valid}$) days. If less than 75% of days are valid, the value is considered missing. | Vegetation |
| nvgt70 | Number of days with exceedance of the dma8epax value above 70 ppb. The value is marked as missing if less than 75% of days contain data. | Human health |
| nvgt100 | Number of days with exceedance of the daily max1h values above 100 ppb. The value is marked as missing if less than 75% of days contain data. | Human health |

# 5   Validating AQ-Bench via machine learning

In this Section, we introduce the AQ-Bench dataset as a machine learning benchmark dataset. This means we combine the data documentation from the previous Section (Sect. 4) with the machine learning task for this dataset. We also provide an evaluation metric, a data split and baseline experiments.

## 5.1 Task description and evaluation metric

The task proposed for the AQ-Bench dataset is to train a machine learning model that maps from metadata in Table 1 to the ozone metric values in Table 2. This can be achieved with individual machine learning algorithms or in one multi-output algorithm.

The evaluation metric for our baselines is the $R^2$, the coefficient of determination,

$$R^2 = 1 - \frac{\sum_{m=1}^{M}(y_m - \hat{y}_m)^2}{\sum_{m=1}^{M}(y_m - \langle y \rangle)^2} \quad \text{with} \quad \langle y \rangle = \frac{1}{M}\sum_{m=1}^{M} y_m \tag{1}$$

where m denotes a sample index, M the total number of samples, $\hat{y}_m$ a predicted output value and $y_m$ a reference target value.

$R^2$ measures the proportion of variance in the output values that the model predicts from the input values. A larger $R^2$ thus denotes a better model and the largest possible value is 1, or 100 %. We choose $R^2$ as it is comparable between all different targets, even if they cover different value ranges. The overall score of the solution is the mean of all scores achieved on the test set for all ozone metrics. For further evaluation of machine learning results, cross validation can be applied. We would like to challenge the machine learning and air pollution researchers to use this rather small dataset as efficiently as possible to extract all inherent information to accurately map onto the ozone metrics.

## 5.2 Data split

We provide a fixed data split within the AQ-Bench dataset to enable a comparison of our baseline results with future solutions, and to provide a suitable data setup for learning (see below). As it is good practice in machine learning, the dataset is split into three subsets for training, validation and hyperparameter tuning, and testing. The three data subsets are required to be independent, while having a similar statistical distribution to prevent the concealment of possible overfitting and an overestimation of accuracy. Because the dataset is relatively small, the split was chosen to be 60/20/20 %, as it is commonly used for datasets of this size. It is indicated in the dataset whether an example belongs to training, validation or test set.

In order to guarantee the spatial independence of the subsets, the data are divided into several spatial zones. The zones were created by spatial clustering, where stations are assigned to the same cluster if they are closer than 50 km (European Union, 2008). Large station clusters were split again into smaller ones to ensure similar statistical distributions of the training, validation and test datasets. The final clusters were randomly assigned to the three datasets. This way, all stations within a spatially dependent cluster are allocated to the same dataset.

## 5.3 Baseline experiments

As baselines for machine learning approaches on the AQ-Bench dataset we present results obtained with three standard machine learning algorithms. For preprocessing, rows with missing values are dropped. Continuous metadata is scaled, each by a quantile range from 25 % to 75 % to avoid influence from outliers. Categorical metadata is one-hot encoded, resulting in 135 input features in total. We drop the *longitude* for our baseline experiments, since this is a circular variable and cannot be used

without additional feature engineering. The preprocessed metadata is called input data in the following. Ozone metrics, which are the targets, are not scaled.

Methods:

- **Linear regression.** Linear regression models the simplest correlation between input and target values. It maps an input data example $\mathbf{x}_m$ with $\hat{y}_m = \mathbf{w}^\mathsf{T} \cdot \mathbf{x}_m + b$, where $\mathbf{w}$ and $b$ are the regression parameters weights and bias. Vector $\mathbf{w} = [w_1, w_2, ..., w_N]^\mathsf{T}$ has the dimension of input vector $\mathbf{x}_m = [x_1, x_2, ..., x_N]^\mathsf{T}$.

- **Neural network.** We train a shallow fully connected neural network with two hidden layers of size 20 and 5 neurons, respectively. We use the Adam optimizer with MSE (mean squared error) loss function, L2 regularization and ReLU (rectified linear unit) as activation function (Goodfellow et al., 2016). Training is performed independently for each ozone metric. We optimized the learning rate and regularization parameter by empirical studies and random search. Through further empirical analyses, we decided on the hyperparameters summarized in Appendix B. The model is written in Tensorflow/Keras (Chollet et al., 2015).

- **Random forest.** Our random forest model (Breiman, 2001) is built with a number of 100 trees for each target, based on empirical studies. As in the case of the neural network, we use the MSE as optimization criterion. We use the RandomForestRegressor of SciKit-learn (Pedregosa et al., 2011).

The baseline results are summarized in Table 3. Comparing the different models, random forest yields the best results for all targets except the *nvgt*-metrics, where the neural network performs best. The linear regression is the worst for all targets except e.g. *75 % percentile* where it is second best after the Random Forest. For some targets, e.g. *average values*, random forest is only slightly better than the neural network. However, there are targets, e.g. *AOT40*, where the gap between the two methods is almost 10 %. The neural network performs best for *nvgt070* and *nvgt100*. The baseline experiment results of *nvgt100* drops in comparison to other targets with partly negative $R^2$-scores. The results of *nvgt070* have the second least scores. These two targets count exceedances of a certain threshold, so that many values equal zero, which might be problematic to capture for standard machine learning algorithms. Except of those, $R^2$ is higher than 50 % for at least one of the three models per target. This shows that there is a quantitative relationship between input data and targets. Nevertheless, for our baseline experiments we used rather simple models, in order to proof the concept. Ozone, as a secondary pollutant with levels highly dependent on the environment and available precursors, is not captured perfectly by these simple baselines.

Table 3: $R^2$-scores of the test set in %. Best results are marked in bold, second best results are underlined.

| Target | Linear regression | Neural network | Random forest |
| --- | --- | --- | --- |
| Average values | 53.69 | 58.25 | **59.75** |
| Daytime average | 55.93 | 56.26 | **62.99** |
| Nighttime average | 49.79 | 56.92 | **59.00** |
| Median | 52.21 | 56.67 | **56.85** |

*(continued on next page)*

*(Table 3 continued from previous page)*

| Target | Linear regression | Neural network | Random forest |
|---|---|---|---|
| 25 % percentile | 52.77 | 56.12 | **62.75** |
| 75 % percentile | 51.75 | 45.92 | **55.65** |
| 90 % percentile | 49.48 | 50.41 | **58.54** |
| 98 % percentile | 47.68 | 54.89 | **59.19** |
| dma8eu | 49.32 | 54.95 | **58.43** |
| avgdma8epax | 54.76 | 58.23 | **62.99** |
| drmdmax1h | 40.21 | 50.12 | **51.53** |
| W90 | 47.90 | 46.15 | **51.29** |
| AOT40 | 45.88 | 50.91 | **59.97** |
| nvgt70 | 26.38 | **31.94** | 30.53 |
| nvgt100 | -32.33 | **12.51** | -66.57 |
| Overall score | 43.03 | **49.35** | 48.19 |
| Overall score (excluding nvgt) | 50.10 | 53.52 | **58.38** |

# 6    Discussion

## 6.1    Opportunities for machine learning in air quality research

With the AQ-Bench dataset, we used our knowledge on environmental influences on ozone, a toxic greenhouse gas, to bundle air quality data and metadata with machine learning approaches. By doing this, we enable a quick entry into machine learning in air quality research on a global scale with reduced machine learning overhead. Our approach enables to use data from various sources that would otherwise be time consuming to acquire and prepare. We provide a ready to use dataset for the machine learning community, to support research on meaningful real-world applications (motivated by Wagstaff, 2012).

One great advantage of using machine learning for air quality research is the possibility to use data from various different sources, especially data which are not directly connected to air pollution via physical or biogeochemical models (e.g. stable nightlights). To explore this opportunity for ozone, we gathered an unprecedented variety of metadata to allow the machine learning approaches to obtain hints on the many interconnected, nonlinear influences, which determine ozone concentrations (see Section 2). As the results from our baseline experiments show, the AQ-Bench dataset bears some potential to exploit these relations with machine learning methods.

Currently not many air pollution researchers use purely data-driven approaches for their studies. With AQ-Bench we offer a first data driven machine learning view on global tropospheric ozone. To achieve the global view, we use the JOIN web interface[4] of the TOAR data center, which provides customized data products from the TOAR database. As proposed by Schultz

---

[4]https://join.fz-juelich.de/

et al. (2020), our approach is to output the demanded metrics directly, and thus to obtain required data products directly from machine learning. Further applications of AQ-Bench could be developed, such as a classification of ozone sites into 'healthy' or 'unhealthy'. Our dataset fits with the vision for benchmark datasets described by Ebert-Uphoff et al. (2017).

## 6.2 Limitations of AQ-Bench

AQ-Bench includes ozone metrics and metadata from 5577 stations and spans a time period of five years. The stations included in AQ-Bench are not distributed equally around the globe. The spatial coverage in most of the regions is low, except in USA, European countries and some regions of East Asia (Japan and South Korea). This raises the question of whether it is possible to generalize machine learning results to regions that are not included in the training data, even if they have similar input metadata. Possibly it may be necessary to use a combination of observational data and numerical models to achieve full global coverage (c.f. Chang et al. (2017)).

Measurement errors, interannual changes and drift result in noisy ozone metrics. Conversely, at least in the current version of AQ-Bench, the input metadata is fixed and has no temporal evolution, an assumption which we can make, because we average over five years of ozone metrics. It cannot be out ruled that within this time major environmental changes could have happened, e.g. settlements could grow or shrink during this time. This means, that metadata as given in AQ-Bench might not be valid for the whole time period of five years. The population density might have increased, the climate zone might have changed, and if a forest was cleared, for example, the land cover would have changed as well. We note that some uncertainty is introduced by the relatively lax requirement of two annual ozone metric values to form a valid 5-year average value (see Appendix B): if both yearly averages correspond to the beginning or to the end of the time period in question, a bias may be introduced if the ozone concentrations exhibit a strong trend, or if the region experienced rapid changes, such as urbanisation.

Another topic is the complexity of the problem, compared to the dataset size. It is doubtful whether simple machine learning models are intricate enough to grasp all complex relationships between ozone and environmental factors. On the other hand, very deep neural networks, which may be capable of learning such patterns, cannot be trained on a dataset with only 5577 samples. In Sect. 5.3 we gave some basic machine learning approaches to find a mapping between the metadata and the target ozone metrics. We assume that the inaccuracies in our baselines partly arise from the complex relationships of ozone with the environment compared to the input dataset size and complexity of these basic machine learning approaches. Furthermore, through a longer aggregation period, we emphasize robust, static features. This aggregation reduces the size of the dataset and makes a global coverage possible. Due to our focus on spatial relationships we consciously ignore time-resolved patterns. We simplify the problem and make machine learning on the dataset easy - but this simplification also comes at the cost of introducing noise and uncertainties. For a more complete description of ozone processes, more input data, additional input variables and time resolved data could be used.

## 6.3 Machine learning challenges arising from AQ-Bench

In order to provide some guidance on how the machine learning results could be improved compared to the standard machine learning methods applied in our baselines (Sect. 5.3), we briefly discuss some techniques here. One aspect to explore is feature

engineering. Currently AQ-Bench includes for example the circular variable *longitude*, which cannot be accessed by the machine learning algorithm without further feature engineering. Other variables could be accumulated, or transformed to improve machine learning results. See e.g. Duboue (2020) for an introduction to the topic. We hope that the research community will be creative in feature engineering.

Another aspect is multi-task learning. The baseline methods were performed independently for each ozone metric, but there may be a connection between them, as they all describe ozone pollution. Therefore, multi-task learning is a promising direction to exploit these connections. See Zhang and Yang (2017) for a review on this topic.

The baseline experiments show that extremes are sparse and thus difficult to catch. For example, the metric *nvgt070* which counts the days where maximum ozone exceeds 70 ppb (which happens at least once a year at approx. 75 % of the stations) gives acceptable results, but *nvgt100* of is not captured well. This is explained by the fact, that there are very few (< 25 %) stations which experience occasionally ozone values above 100 ppb. Extremes can be captured by imbalanced learning. See He and Garcia (2009) for a review on learning from imbalanced data.

## 7  Data and code availability

The AQ-Bench dataset is available in .csv format at https://doi.org/10.23728/b2share.30d42b5a87344e82855a486bf2123e9f (Betancourt et al., 2020). To enable a machine learning quick start on the AQ-Bench dataset with reproduction of the baseline experiments, we also provide an introductory jupyter notebook on https://gitlab.version.fz-juelich.de/esde/machine-learning/aq-bench. To start it directly in your browser, click the button "launch on binder" in the readme of this repository.

## 8  Conclusions

In this paper, we introduced AQ-Bench as a benchmark dataset for machine learning on global air quality metrics. It allows to explore different machine learning methods on the real-world problem of air quality analyses. Specifically, the machine learning task is to map station metadata to air quality metrics at 5577 measurement stations around the globe and to optimize the results with hyperparameter tuning and data engineering. The usability of the dataset is documented through the results from our three baseline machine learning solutions. These methods show robust relations between the input data (geospatial features) and the targets (ozone metrics), and these relations are understandable from an atmospheric chemistry point of view. As data driven techniques for air quality research are emerging, we present a first benchmark dataset on the global scale. The purpose and significance of AQ-Bench is twofold: first, it has never been tried before to exploit a rich collection of geospatial datasets to find out which fraction of ozone pollution can be attributed to such more or less static geographical features. Second, this problem definition makes some low-level air quality analysis easily accessible to data scientists with little or no background in atmospheric chemistry. Following the vision of Ebert-Uphoff et al. (2017) to design benchmarks that bridge geoscience and data science, the key features of AQ-Bench are:

– **Active research area:** Ozone is a highly relevant and active field of research, as it harms living beings and the ecosystem. Ozone research benefits from making data available and developing data driven methods for ozone assessment.

– **Understandable context:** We introduced the complex mechanisms behind ozone formation as well as physical and chemical processes in Sect. 2, to make the scientific context of this dataset understandable to everyone, even without prior knowledge.

– **Impact on data science:** Since AQ-Bench is relatively small and thus easy to handle, it is suitable for beginners in programming. AQ-Bench can be trained in less than a minute on a common personal computer without GPUs, so one can quickly iterate through different algorithms and configurations. Yet noise, the small size of the data set and the complicated underlying processes make it challenging to achieve satisfactory machine learning results on this dataset.

– **A means to evaluate success:** We propose $R^2$, the coefficient of determination, as an evaluation metric for AQ-Bench.
It is a suitable metric because it measures the proportion of variance in the output values that the model predicts from the input values. It is comparable between all targets.

– **Quick start:** To start machine learning on AQ-Bench in a common browser, launch the "binder" in the following Git repository: https://gitlab.version.fz-juelich.de/esde/machine-learning/aq-bench. Running the introductory notebook on binder enables users to try out different training algorithms and hyperparameters directly in the browser.

– **Citability and reproducibility:** The dataset has a DOI, and the baseline experiments can be reproduced with the code that is openly available on Github (see Sect. 7).

We hope that the AQ-Bench dataset will help to advance data driven techniques in the field of air quality research, and form the basis for future experiments and research.

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

**Appendix A: Technical details on the station metadata of the AQ-Bench dataset**

Table A1: Technical details on the station metadata of the AQ-Bench dataset, updated from Schultz et al. (2017). Please note that in order to keep this table uncluttered, we have summarized all types of land cover in 25 km area, all population density and all nightlight variables in one row each.

| Variable | Data source | Reference |
| --- | --- | --- |
| Country | Information given by data providers | |
| HTAP region | Derived from gridded data: TIER 1 regions from the Task Force on Hemispheric Transport of Air Pollution with an original resolution of 0.1° | Koffi et al. (2016) |
| Climatic zone | Derived from gridded data: IPCC 2006 classification scheme for default climate regions with a resolution of 5' | https://esdac.jrc.ec.europa.eu/ projects/RenewableEnergy/, accessed 23 Mar 2021 |
| Longitude | Information given by data providers. Quality controlled by TOAR database administrators | |
| Latitude | Information given by data providers. Quality controlled by TOAR database administrators | |
| Altitude | Information given by data providers. Quality controlled by TOAR database administrators | |
| Relative altitude | Derived from the ETOPO 1 digital elevation model and the station altitude | Amante and Eakins (2009) |
| Type | Information given by data providers | |
| Type of area | Information given by data providers | |
| Landcover in 25km area | Derived from gridded data: Yearly land cover type L3 from the MODIS MD12C1 collection with an original resolution of 0.05°. The year 2012 and the IGBP classification scheme were used | https://ladsweb.modaps.eosdis.nasa.gov/ missions-and-measurements/ products/MCD12C1/, accessed 23 Mar 2021 |
| Wheat production | Derived from gridded data: annual wheat production of the year 2000 according to the Global Agro-Ecological Zones data, version 3 with an original resolution of 5' | www.fao.org/, accessed 23 Mar 2021 |
| Rice production | Derived from gridded data: annual rice production of the year 2000 according to the Global Agro-Ecological Zones data, version 3 with an original resolution of 5' | www.fao.org/, accessed 23 Mar 2021 |

*(continued on next page)*

| Variable | Data source | Reference |
| --- | --- | --- |
| $NO_x$ emissions | Derived from gridded data: annual $NO_x$ emissions of the year 2010 from EDGAR HTAP inventory V2 with an original resolution of 0.1° | Janssens-Maenhout et al. (2015) |
| $NO_2$ full column | Derived from gridded data: 5-year average (2011-2015) tropospheric $NO_2$ column value from the Ozone Monitoring Instrument (OMI) instrument on NASA AURA with an original resolution of 0.1° | Krotkov et al. (2016) |
| Population density | Derived from gridded data: GPWv3 population density of the year 2010 with an original resolution of 2.5' | CIESIN (2005) |
| Nightlight | Derived from gridded data: stable nighttime lights of the year 2013 extracted from ghe NOAA DMSP product with an original resolution of 0.925 km | https://ngdc.noaa.gov/eog/dmsp/ downloadV4composites.html, accessed 23 Mar 2021 |

## Appendix B: Data capture criteria

The data capture criteria applied in this work ensure robustness of the ozone metrics. Data capture criteria of hourly to annual metrics are applied through the JOIN web service (https://join.fz-juelich.de/), as described in Schultz et al. (2017). The 5-year mean and its data capture criterion were applied in this work. One exception is the *average values* metric which does not have a data capture criterion in JOIN. Here we have verified that more than 2200 hourly values are processed to calculate the metric, and that the average hourly data capture of all stations is above 50 %. The flowchart in Fig. B1 shows an example data capture criterion as applied in the AQ-Bench dataset. All data capture criteria are summarized in Table 2 of this work.

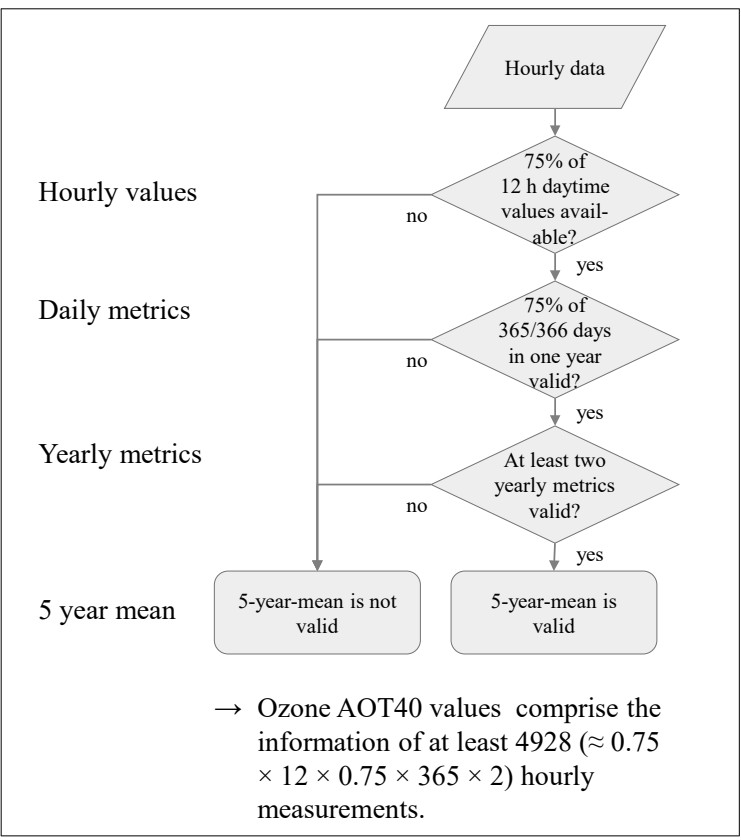

**Figure B1.** Data capture criteria for the *AOT40* metric

 **Appendix C: Data editing**

Some data from TOAR/JOIN were modified in order to make it more understandable and user friendly.

– *HTAP region* was updated according to the number code:

| # | Replaced with | Description |
|---|---|---|
| 2 | OCN | Non-arctic/Antarctic Ocean |
| 3 | NAM | US + Canada (up to 66° N; polar circle) |
| 4 | EUR | Western + Eastern EU + Turkey (up to 66° N; polar circle) |
| 5 | SAS | South Asia: India, Nepal, Pakistan, Afghanistan, Bangladesh, Sri Lanka |
| 6 | EAS | East Asia: China, Korea, Japan |
| 7 | SEA | South East Asia |
| 8 | PAN | Pacific, Australia + New Zealand |
| 9 | NAF | Northern Africa + Sahara + Sahel |
| 10 | SAF | Sub Saharan / sub Sahel Africa |
| 11 | MDE | Middle East: S. Arabia, Oman, Iran, Iraq etc. |
| 12 | MCA | Mexico, Central America, Caribbean, Guyana, Venezuela, Columbia |
| 13 | SAM | South America |
| 14 | RBU | Russia, Belarus, Ukraine |
| 15 | CAS | Central Asia |
| 16 | NPO | Arctic Circle (North of 66° N) + Greenland |
| 17 | SPO | Antarctic |

– *Climatic zone* was updated according to the number code:

| # | Replaced with |
|---|---|
| 1 | warm_moist |
| 2 | warm_dry |
| 3 | cool_moist |
| 4 | cool_dry |
| 5 | polar_moist |
| 6 | polar_dry |
| 7 | boreal_moist |

*(Table continued on next page)*

| # | Replaced with |
|---|---|
| 8 | boreal_dry |
| 9 | tropical_montane |
| 10 | tropical_wet |
| 11 | tropical_moist |
| 12 | tropical_dry |

– The variable *type* was harmonized, as there are some types which appear only once or twice. These types were replaced with the category they go best with:
  - "agricultural", "commercial", "other-agricultural", "other-marine" were replaced with "other"
  - "rural" was replaced with "background"
  - "urban" was replaced with "unknown".

  Five types remain: "background", "industrial", "traffic", "other" and "unknown".

– The variable *type_of_area* was harmonized in the same way as *type*:
  - "alpine grasslands", "background", "forest" and "marine" were replaced with "unknown"
  - "rural-nearcity" and "rural-regional" were replaced with "rural"
  - "rural-remote" was replaced with "remote"
  - "Urban" was replaced with "urban".

  Five types of area remain: "rural", "urban", "suburban", "remote" and "unknown".

– The station with id 4587 was sorted out, because it was a remote background station in Romania which reported one of the highest *o3_average value* of all stations (65.5899 ppb), and had a low data coverage. We suspect these values are faulty.

– The station with id 4589 was sorted out because it reported a *max_population_density_5km* of ca. 1 million per square kilometer which we suspect is faulty.

# Appendix D: Hyperparameters for baselines

**Table D1.** Hyperparameters for the neural network training in Sect. 5.3. They are determined from empirical studies and random search.

| Target | Learning rate | L2 lambda | Batch size | Epochs |
|---|---|---|---|---|
| Average values | 1.0E-04 | 1.0E-02 | 32 | 250 |
| Daytime average | 1.0E-04 | 1.0E-02 | 32 | 250 |
| Nighttime average | 1.0E-04 | 1.0E-02 | 32 | 250 |
| Median | 1.0E-04 | 1.0E-02 | 32 | 250 |
| 25 % percentile | 1.0E-03 | 1.0E-02 | 64 | 100 |
| 75 % percentile | 1.0E-03 | 1.0E-02 | 256 | 250 |
| 90 % percentile | 1.0E-03 | 1.0E-02 | 256 | 250 |
| 98 % percentile | 1.0E-03 | 1.0E-02 | 256 | 250 |
| dma8eu | 1.0E-03 | 1.0E-02 | 128 | 250 |
| avgdma8epax | 1.0E-04 | 1.0E-02 | 32 | 250 |
| drmdmax1h | 2.0E-04 | 1.0E-02 | 32 | 150 |
| W90 | 1.0E-04 | 1.0E-02 | 32 | 250 |
| AOT40 | 1.0E-02 | 1.0E-02 | 128 | 250 |
| nvgt070 | 1.0E-04 | 1.0E-02 | 32 | 150 |
| nvgt100 | 1.0E-05 | 1.0E-02 | 32 | 200 |

*Author contributions.* CB and TS prepared the dataset, developed the software and conducted the baseline experiments. CB, SS and TS prepared the initial manuscript draft. RR and MGS reviewed and edited the manuscript. All authors read and approved the manuscript. RR and MGS supervised the project. CB coordinated the project.

*Competing interests.* MGS is topic editor of the ESSD Journal.

*Acknowledgements.* CB, SS and MGS acknowledge funding from ERC-2017-ADG#787576 (IntelliAQ). TS and MGS are partly funded through the program Supercomputing & Big Data of the Helmholtz Association's research field Key Technologies. The authors gratefully acknowledge the computing resources granted by Jülich Supercomputing Centre (JSC). The Graphical Abstract of this paper was designed with icons from Flaticon (https://www.flaticon.com/). We gratefully acknowledge the comments and suggestions of two anonymous reviewers and the topical editor.