# Peer review of "AQ-Bench: A Benchmark Dataset for Machine Learning on Global Air Quality Metrics"

_Earth System Science Data, 2020_

## Referee Comment (RC1)

This study introduced a AQ-Bench dataset for machine learning. This dataset includes aggregated air quality data of more than 5500 air quality monitoring stations originated from Tropospheric Ozone Assessment Report (TOAR). This is an interesting dataset that contains both data of tropospheric ozone and data of many ozone-related factors, which enables a quick start of machine learning with the AQ-Bench dataset. I would suggest publication after the following issues are addressed.

1. My main concern is that this dataset is a five-year averaged ozone data and does not include any time series information of ozone in the final data product. An average over such a long time would filter out a lot of important information. I am afraid that such a small dataset may limit the applications for many machine learning methods to investigate the complex and nonlinear relationship between ozone and other factors. Moreover, because there is no time series information, it also makes it difficult to well verify the trained model. For example, can the model built using these 5 years of data be used for another period? Therefore, even for climate-scale analysis, I recommend adding some time series data to this dataset.

2. More efforts are needed to show the quality control processes and the error estimation of AQ-Bench dataset. Although it have been stated that the data originates from the TOAR database, I think as a data paper, it is necessary to provide more complete information about data sources, quality control and data accuracy. For example, how many observation samples were used for the 5-year averaging at different sites? Are the emission data reference to a base year or the average of the five years?

3. The AQ-Bench dataset was validated through using three machine learning methods with a defined task. As a machine learning dataset, I think the first step is to verify the reliability of the data itself and then its usability in machine learning. For the machine learning tests, how to judge the quality of the dataset based on the training results? Is there a well-defined standard for that? Additionally, more explanation are needed for the purpose and significance of the task mapping from metadata in Table 1 to the ozone metric values in Table 2.

4. The performance of a machine learning method depends on the configured parameters of the method in the experiment. The manuscript shows the configurations (e.g., 100 trees in RF) of different method in the baseline experiments, but it does not explain the reason for adopting such configurations. At least, some discussions on this issue are needed.

5. Table 3, please clarify if this is the result for the validation datasets?

---

## Author Comment (AC1)

**Response to the anonymous Referee #1 Comments on "AQ-Bench: A Benchmark Dataset for Machine Learning on Global Air Quality Metrics" by Betancourt et al.**

This study introduced a AQ-Bench dataset for machine learning. This dataset includes aggregated air quality data of more than 5500 air quality monitoring stations originated from Tropospheric Ozone Assessment Report (TOAR). This is an interesting dataset that contains both data of tropospheric ozone and data of many ozone-related factors, which enables a quick start of machine learning with the AQ-Bench dataset. I would suggest publication after the following issues are addressed.

**Answer**: We appreciate the comments and suggestions from Referee #1 and are grateful for the opportunity to improve our study. All questions and comments are addressed individually below. Since some comments cover multiple aspects, we have split them into subsections (a), (b) ...

1. My main concern is that this dataset is a five-year averaged ozone data and does not include any time series information of ozone in the final data product. (a) An average over such a long time would filter out a lot of important information. (b) I am afraid that such a small dataset may limit the applications for many machine learning methods to investigate the complex and nonlinear relationship between ozone and other factors. (c) Moreover, because there is no time series information, it also makes it difficult to well verify the trained model. For example, can the model built using these 5 years of data be used for another period? Therefore, even for climate-scale analysis, I recommend adding some time series data to this dataset.

**Answer**:

AQ-Bench has been designed as a low-barrier entry dataset for atmospheric scientists wanting to make first experiences with machine learning techniques, and machine learners who want to familiarize themselves with atmospheric data. It therefore focuses on the analysis of spatial patterns. Of course, the reviewer is right that ozone levels are influenced by time-dependent phenomena, even on longer scales, and AQ-Bench can therefore only capture a snapshot of the relations between geospatial data and multi-annual ozone metrics. However, we believe that there are enough interesting aspects in the dataset as it is, and the addition of time-varying features would complicate the analysis and add extra computational burden to the machine learning solutions. Furthermore, there are many published studies on time series analyses with machine learning techniques, and they sometimes come with easily accessible code and data (e.g. Kleinert et al., 2020 as cited in our manuscript). We are developing the air quality benchmark dataset concept further and are planning to produce a benchmark dataset for learning spatiotemporal relations on much larger scales at a later stage.

(a): Filtering out information in the time domain is a valid approach when studying environmental relationships on longer time scales. The metrics available in JOIN can be queried over the time span of one year, as it makes physical/scientific sense to consider one value that is representative for the ozone distribution at a site over this period. Averaging over five years makes the values more robust to drift and interannual changes. By filtering out the time resolved information, we emphasize robust, static features, which can be analyzed on the global scale. We added the sentence 'Furthermore, through a longer aggregation period, we emphasize robust, static features. This

aggregation reduces the size of the dataset and makes a global coverage possible. Due to our focus on spatial relationships we consciously ignore time-resolved patterns.' in the limitations part (Section 6.2, line 331ff in the marked-up manuscript).

(b): AQ-Bench does not aim to enable the exploration of "[all] complex and nonlinear relationship(s) between ozone and other factors", but instead defines one specific research task which can be addressed with a variety of machine learning techniques. We understand that our ambitions could have been overstated a bit in the abstract and introduction and therefore added the sentence 'The purpose of this dataset is to produce estimates of various long-term ozone metrics based on time-independent local site conditions.' in the abstract, line 8f of the marked-up manuscript. We furthermore added the sentences 'It is doubtful whether simple machine learning models are intricate enough to grasp all complex relationships between ozone and environmental factors. On the other hand, very deep neural networks, which may be capable of learning such patterns, cannot be trained on a dataset with only 5577 samples.' in the limits part (Section 6.2, line 325ff in the marked-up manuscript).

(c): The reviewer raises the valid point of generalisability here, which is, however, not in scope of AQ-Bench. The target of AQ-Bench is to obtain the best possible coefficient of determination for various ozone metrics in relation to the predictor variables. This requires no further verification as such. Of course it would be interesting to broaden the task to include temporal changes in predictor variables, but this has to be the subject of a different study.

> 2. More efforts are needed to show the quality control processes and the error estimation of AQ-Bench dataset. Although it have been stated that the data originates from the TOAR database, I think as a data paper, it is necessary to provide more complete information about data sources, quality control and data accuracy. (a) For example, how many observation samples were used for the 5-year averaging at different sites? (b) Are the emission data reference to a base year or the average of the five years?

**Answer**:

Thank you very much for drawing our attention to this point. In general, the quality control of ozone data in TOAR is in the hand of the air quality networks, and data as well as metadata were also quality controlled by TOAR (Schultz et al, 2017, as cited in our paper). We have added a couple of sentences regarding quality control in TOAR in Section 4, where TOAR data products are introduced. More specifically, line 156ff in the marked-up manuscript now reads 'The data providers conduct quality control on these data by calibrating the measurement devices and setting suitable instrument parameters. In a second step of data curation, the TOAR database administrators conduct a statistical analysis of the data to identify and remove low-quality data (Schultz et al., 2017).'. Line 177ff reads 'Some data, for instance station coordinates and altitude are given by the data providers and quality controlled by TOAR. Others were derived from data sources with individual quality control, such as satellite earth observations.'

(a) We agree that more clarity is needed regarding the data entering the AQ Bench summary statistics. However, we see relatively little value in a long table with the absolute number of samples included. There are two reasons: First, we apply data capture criteria which are established in the ozone research community. When these criteria are met, the data is considered statistically robust and thus valid. We do not think more details are needed for the users of a benchmark dataset. Second, even if we would give the data completeness of every data point, it is still an open field of

research how uncertainty induced by missing data propagates through metrics calculation (Section 3.3 of Lefohn et al., 2018, as cited in our manuscript). Therefore, we are of the opinion that such a table is not needed by the user. Instead we have added a reference to the data capture criteria in Table 2, which were used in compiling our dataset in Section 4.2, line 224f of the marked-up manuscript 'A summary of all metrics and their data capture criteria is given in Table 2.', to clarify the data capture process. We have also added more details on percental data capture of hourly ozone and a graph to clarify the data capture criteria of an exemplary ozone metric (*ozone_aot40*) in Appendix B. We reference this Appendix in the dataset description part, in Section 4.2, line 224f of the marked-up manuscript.

(b) We added a table in the Appendix A where more information on the geospatial data is given, including the year, and the data source. We reference this appendix in the part where the metrics of the datasets are described, in Section 4.1, line 202f of the marked-up manuscript.

3. The AQ-Bench dataset was validated through using three machine learning methods with a defined task. (a) As a machine learning dataset, I think the first step is to verify the reliability of the data itself and then its usability in machine learning. (b) For the machine learning tests, how to judge the quality of the dataset based on the training results? Is there a well-defined standard for that? (c) Additionally, more explanation are needed for the purpose and significance of the task mapping from metadata in Table 1 to the ozone metric values in Table 2.

**Answer**:
We assessed the issue of verifying the reliability and quality of our data carefully. We hope that our answers and changes regarding remark 2 have made this clearer.

(a) The data used in AQ-Bench is reliable because we are using TOAR data which was already verified, and has been the basis for many studies, e.g. the Tropospheric Ozone Assessment Report (TOAR, as cited in our manuscript). We hope that the reference to TOAR and our corrections in regard to remark 1 suffice as a verification for the reliability of the data. Regarding the usability, we have added a summary in the conclusion, Section 8, line 363ff of the marked-up manuscript: 'The usability of the dataset is documented through the results from our three baseline machine learning solutions. These methods show robust relations between the input data (geospatial features) and the targets (ozone metrics), and these relations are understandable from an atmospheric chemistry point of view.'.

(b) For a first estimation of the suitability of the data for machine learning, it is sufficient to use a standard method and a suitable evaluation metric with a standard train-validation-test split (Section 5). The metric for success is clearly defined as it is common practice in machine learning applications. We chose to use the R2 score which has to be larger than zero. We mention up-front that the dataset is validated by the machine learning baselines: 'Baseline scores obtained from a linear regression method, a fully connected neural network and random forest are provided for reference and validation.', abstract, line 10f of the marked-up manuscript. We mention that besides these first baselines, the dataset and machine learning results can be further validated by cross-validation: 'For further evaluation of machine learning results, cross validation can be applied.', Section 5.1, line 242 of the marked-up manuscript.

(c) We added a detailed explanation in the discussion on the purpose and significance of AQ-Bench in, Section 8, line 367ff of the marked-up manuscript: 'The purpose and significance of AQ-Bench is twofold: first, it has never been tried before to exploit a rich collection of geospatial datasets to find out which fraction of ozone pollution can be attributed to such more or less static geographical features. Second, this problem definition makes some low-level air quality analysis easily accessible to data scientists with little or no background in atmospheric chemistry.' In this sense, we liken AQ-Bench to the famous *MNIST* dataset, which played a major role in the early stages of the development of machine learning techniques for image classification. The AQ Bench dataset is simple, but it still highlights several specific challenges of atmospheric data analysis, which differ from most other classical machine learning problems (cf. Schultz et al. (2020) as cited in our manuscript). We added one additional remark on the purpose of the dataset in the abstract (line 8f of the marked-up manuscript): 'The purpose of this dataset is to produce estimates of various long-term ozone metrics based on time-independent local site conditions.'.

4. The performance of a machine learning method depends on the configured parameters of the method in the experiment. The manuscript shows the configurations (e.g., 100 trees in RF) of different method in the baseline experiments, but it does not explain the reason for adopting such configurations. At least, some discussions on this issue are needed.

**Answer**:

It is common to use standard machine learning algorithms (the so called 'vanilla' methods) with hyperparameters suitable for the given task as baselines for a benchmark dataset. This is done to show that the dataset is suitable for machine learning. The choice of machine learning methods and optimization of hyperparameters is then left open to the users. For the random forest we used 100 trees. This is a common choice producing good results on the AQ-Bench dataset. For the neural network, we chose a shallow architecture based on our empirical studies. The reported learning rate and L2 regularization parameter were established through random search. We added more justification and explanation on hyperparameters. In the baseline section (Section 5.3), we rewrote two sentences. Line 272f of the marked-up manuscript now reads: 'We optimized the learning rate and regularization parameter by empirical studies and random search. Through further empirical analyses, we decided on the hyperparameters summarized in Appendix B.'. Line 275f of the marked-up manuscript now reads: 'Our random forest model (Breiman, 2001) is built with a number of 100 trees for each target, based on empirical studies. As in the case of the neural network, we use the MSE as optimisation criterion.' In Appendix D, line 620f of the marked-up manuscript, we added the sentence 'They are determined from empirical studies and random search. 'In the conclusion (Section 8), line 361f of the marked-up manuscript, we added: 'Specifically, the machine learning task is to map station metadata to air quality metrics at 5577 measurement stations around the globe and to optimize the results with hyperparameter tuning and data engineering.'

5. Table 3, please clarify if this is the result for the validation datasets?

**Answer**:
Table 3 shows the results on the test set. To clarify this, we changed the caption of Table 3 to: 'R2-scores of the test set in %. [...]'.

---

## Author Comment (AC2)

**Response to the anonymous Referee Comment #2 on "AQ-Bench: A Benchmark Dataset for Machine Learning on Global Air Quality Metrics" by Betancourt et al.**

**General Comments**

This paper has the primary goal of presenting a dataset combining several long-term metrics related to surface ozone concentration together with explanatory parameters (metadata) related to the ozone measurement stations, which can be used as inputs to a machine learning framework to predict the metrics. It also presents a benchmark application of several simple machine learning algorithms to this problem as an illustrative example. Overall the dataset and its goal are very useful, but I believe several improvements should be made prior to its publication.

**Answer**: We want to thank the Referee #2 for the valuable comments and questions. We appreciate the opportunity to improve our study. In the following, we address each of the comments individually.

The goal (as summarized best, I think, at the start of Section 4) is to produce estimates of various long-term ozone metrics based on local site conditions, rather than (for example) short-term estimates based on atmospheric characteristics (temperature, wind, etc.). I think this should be more clearly and explicitly stated up-front, ideally in the abstract, so the specific goals for which this dataset has been created are more immediately clear to a reader.

**Answer**:
We added a sentence in the abstract (line 8f in the marked-up manuscript), to clarify the goal of this dataset: 'The purpose of this dataset is to produce estimates of various long-term ozone metrics based on time-independent local site conditions. We combine this task with a suitable evaluation metric.'.

Although it is perhaps beyond the scope of this work, I believe that the introduction should also make mention of the use of machine learning in other areas related to atmospheric chemistry and air quality. Machine learning techniques are often used as emulators or surrogate models for more computationally complex components of atmospheric chemistry models, i.e., to replace complicated and costly atmospheric chemistry calculations with simpler machine learning surrogates to improve computational performance of the models. Second, machine learning is being extensively used in the calibration of low-cost sensors for air quality, in order to account for the many sources of interference with the measurements of these sensors and allow them to more effectively supplement the existing monitoring networks for pollutants such as ozone. Although such use of machine learning is beyond the scope of the present work, I believe it is still worthwhile to draw attention to these other areas of ongoing work combining machine learning and atmospheric science.

**Answer**:
Thank you. These applications of machine learning in atmospheric chemistry are certainly interesting and worth to mention. We added two sentences in the introduction pointing to studies on parameterization and low cost sensors: 'Moreover, within computationally complex components of atmospheric chemistry models, machine learning techniques are used as emulators or surrogate models. They replace for example costly atmospheric chemistry and micro-physical calculations to improve computational performance of the models (e.g. Kelp et al., 2020). In addition, machine learning is applied in the calibration of low-cost sensors for air quality measurements in order to account for the diverse sources of interference with these measurements (e.g. Schmitz et al., 2021; Wang et al., 2020).' (Section 1, line 40ff of the marked-up manuscript).

> In Table 1, sources should be provided for these metadata, especially data on NOx emissions, NOx column, and night light intensity. It should also be made more clear what time period these represent, i.e., are they the average over the entire time interval, or more representative of the situation in one particular year. I suspect that the answer will vary by dataset; for example, population density might be based on census data from a particular year, while NOx column density might be derived from a satellite, and therefore represent an average (although only for clear-sky periods during which the satellite is overhead) over many years. Details do not necessarily need to be provided for each underlying dataset, but the initial source for each data should be made clear so the reader can look into the details.

**Answer**:
As referee #2 assumed correctly, the answer varies by dataset. We added a table in the Appendix A where more information on the geospatial data is given, including the data sources for more background information if needed. This table also contains links to the data documentation, which provides further information for the interested reader. We reference this appendix in Section 4.1 where the metrics of the datasets are described, line 202f of the marked-up manuscript.

> In Table 2, there seems to be an inconsistent definition for data completeness applied across these metrics (e.g., the overall average requires only one valid datapoint, percentiles require 10, and daytime and nighttime averages require 75% completeness). I would suggest, at a minimum, that a "completeness" metric also be included in the data set, indicating what fraction of the expected total number of hourly measurements for each site were actually present in the dataset. This would allow users to potentially make different decisions about what values they consider to be "valid" to their applications. Furthermore, if the datasets are more complete towards the beginning or end of the time period in question, this could combine with the fact that the underlying metadata (e.g., population density) might be changing over time, and be a confounding factor in the performance of the machine learning approaches. While this is a complex issue to address, some mention should be made of this limitation on the data which a single "completeness" metric will not capture.

**Answer**:
We agree that more clarity is needed regarding the data entering the AQ Bench summary statistics. However, we see relatively little value in a long table with the absolute number of samples included. There are two reasons: First, we apply data capture criteria which are established in the ozone research community. When these criteria are met, the data is considered statistically robust and

thus valid. We do not think more details are needed for the users of a benchmark dataset. Second, even if we would give the data completeness of every data point, it is still an open field of research how uncertainty induced by missing data propagates through metrics calculation (Section 3.3 of Lefohn et al., 2018, as cited in our manuscript). Therefore, we are of the opinion that such a table is not needed by the user. Instead we have added a reference to the data capture criteria in Table 2, which were used in compiling our dataset in Section 4.2, line 224f of the marked-up manuscript 'A summary of all metrics and their data capture criteria is given in Table 2.', to clarify the data capture process. We have also added more details on percental data capture of *ozone_average_values* and a graph to clarify the data capture criteria of an exemplary ozone metric (*ozone_aot40*) in Appendix B. We reference this Appendix in the dataset description part, in Section 4.2, line 224f of the marked-up manuscript.

Concerning the potential source of errors for unequally sampled data we added some discussion on this in Section 6.2: 'We note that some uncertainty is introduced by the relatively lax requirement of two annual ozone metric values to form a valid 5-year average value (see Appendix B): if both yearly averages correspond to the beginning or to the end of the time period in question, a bias may be introduced if the ozone concentrations exhibit a strong trend, or if the region experienced rapid changes, such as urbanisation.' (line 320ff in the marked-up manuscript).

> In section 5.1, a particular machine learning task and evaluation metric is introduced. While this is useful for benchmarking purposes, it may not be appropriate in all cases, depending on the application. For example, determination of whether metrics fall into certain discrete classification regimes, e.g., "healthy" or "unhealthy", may be a desired goal in certain applications. While it is impossible to account for all potential uses of these metrics, some mention should be made that this is only one of many possible machine learning goals and evaluation metrics.

**Answer**:
Thank you for these suggestion. We added a sentence to the Conclusions which reads: "Further applications of AQ-Bench could be developed such as a classification of ozone sites into 'healthy' or 'unhealthy'."

> **Specific Comments**

> Line 36: O3 and Ozone are both used; this may be redundant.

**Answer**:
Yes, 'O$_3$' is redundant. Section 1, line 37f of the marked up manuscript now reads: 'The input data are directly mapped to a specific data product, e.g. from meteorological and past ozone measurements to the next day's maximum ozone value.'.

> Line 79: Starting the sentence with "I.e." seems awkward to me; I would suggest rephrasing this.

**Answer**:

We rephrased this sentence. Section 2.1, line 85f of the marked up manuscript now reads: 'Agricultural fields, forests, and grasslands therefore yield different magnitudes and seasonal cycles of VOC emissions (Simpson et al. 1999).'.

Lines 105-106: suggest revising to "Ozone irreversibly damages plant tissue when the plant leaves take it up (Schraudner et al., 1997) leading to reduced crop yields (Mills et al., 2011)."

**Answer**:

We applied the correction as suggested by referee #2 (Section 2.3, line 114f).

Line 110: suggest removing "exemplary".

**Answer**:

We removed "exemplary" (Section 2.4, line 118f).

Lines 127-128: This sentence is awkward; I am not sure what the correct phrasing should be, but the authors should consider revising this sentence.

**Answer**:

We rephrased the sentence and hope that it is now clear for the reader. Section 2.4, line 136ff of the marked up manuscript now reads: 'The "radius of influence" within which ozone is determined by nearby precursor emissions and deposition surfaces is typically about 25 km in mid-latitude areas (European Union, 2008).'.

Line 132: suggest revising "ground-level ozone levels" to "ground-level ozone concentrations" to avoid the repeated word.

**Answer**:

We revised the sentence as suggested (Section 2.4, line 142).

Line 169: It is unclear whether "mean ozone metrics" refers to the average values of different metrics across the given time period, or simply the average ozone concentrations in different locations. This may need to be clarified.

**Answer**:

We rephrased the sentence as follows: Section 4, line 184f of the marked up manuscript now reads: 'The AQ-Bench dataset consists of metadata and aggregated ozone metrics from the years 2010-2014 at 5577 measurement stations all over the world, compiled from the TOAR database'. We scanned the manuscript for other occurrences of this ambiguity, and rephrased them where necessary. Section 3.1, line 166 now reads '[...] also basic statistics such as average, median and percentiles are available [...].'.

Lines 233-236: It is unclear how these clusters are used. Are entire clusters assigned to one of the three data subsets, or are individual members of the clusters divided between subsets, such that each subset will get at least one sample from each cluster? Also, how is

the issue of sparse measurements in South America or Africa, for example, addressed using this spatial clustering approach?

**Answer**:

Clustering was used to prevent putting strongly correlated measurements from neighboring stations into different datasets as this would lead to overfitting and overly optimistic evaluation of the machine learning results. This is why we always sort one cluster into one of the three datasets. Stations in sparsely covered regions will generally not be clustered as their distance is usually larger than the 50 km applied as selection criterion. Section 5.2, line 252f of the marked-up manuscript: 'In order to guarantee the spatial independence of the subsets, the data are divided into several spatial zones. The zones were created by spatial clustering, where stations are assigned to the same cluster if they are closer than 50 km (European Union, 2008). Large station clusters were split again into smaller ones to ensure similar statistical distributions of the training, validation and test datasets. The final clusters were randomly assigned to the three datasets. This way, all stations within a spatially dependent cluster are allocated to the same dataset.' We hope this explanation is sufficient to understand our approach.